



# Zonal variations of the vertical distribution of atmospheric aerosols over the Indian region and the consequent radiative effects

Nair K. Kala[1,2], Narayana Sarma Anand[2], Mohanan R. Manoj[2], Harshavardhana S. Pathak[2], Krishnaswamy K. Moorthy[2] and Sreedharan K. Satheesh[1, 2, 3]

[1]Centre for Atmospheric and Oceanic Sciences, Indian Institute of Science, Bengaluru, India
[2]Divecha Centre for Climate Change, Indian Institute of Science, Bengaluru, India
[3]DST-Centre of Excellence in Climate Change, Indian Institute of Science, Bengaluru, India

Correspondence: Nair K. Kala (kalanair56@gmail.com)

**Abstract.** The vertical structure of atmospheric aerosols over the Indian mainland and the surrounding oceans and its spatial
distinctiveness are characterized using long-term (2007 – 2020) spaceborne lidar observations, satellite-retrieved aerosol
optical depths and assimilated aerosol single scattering albedo. The consequence of these on the spatial distribution of aerosol-
induced atmospheric heating is estimated using radiative transfer calculations. The results show strong, seasonally varying
zonal gradients in the concentrations and vertical extent of aerosols over the study region. In general, while over the oceans,
aerosol concentrations decrease rather monotonically with increase in altitude (from its highest value near the surface), over
the mainland, the concentrations initially increase from the surface to about 1 km before decreasing towards higher altitudes,
in all seasons over Central India and during summer monsoon season in northern India. This is attributed to the seasonal
variations in the source strengths and the atmospheric boundary layer dynamics. Compared to the surrounding oceans, where
the vertical extent of aerosols is confined within 3 km, the aerosol extinction coefficients extend to considerably higher altitudes
over the mainland, reaching as high as 6 km during pre-monsoon and monsoon seasons. Longitudinally, the vertical extent is
highest around 75 °E and decreasing gradually on either side over the peninsular India. In the west, the concentrations and
vertical extent of aerosols are highest during summer/monsoon due to the lofting and strong advection of mineral dust and sea
salt aerosols. Particulate depolarization ratio profiles affirm the ubiquity of dust aerosols in western India during monsoon.
Dust aerosols are distributed all the way from surface to 6 km over the north-western semi-arid regions. While the presence of
low-altitude dust aerosols decreases further east, the high-altitude (above 4 km) dust layers are observed to remain aloft
throughout the year with seasonal variations in its zonal distribution over north-western India. Southern peninsular India and
its surrounding oceans are marked with high-altitude (around 4 km) dust aerosols during the monsoon season. Radiative
transfer calculations show that these changes in vertical distribution of aerosol loading and types result in enhanced
atmospheric heating at the lower altitudes during pre-monsoon, with prominent heating within 2 – 3 km throughout the Indian
region. These results will have large implications for aerosol-radiation interactions in regional climate simulations.



# 1 Introduction

Atmospheric aerosols are known to regulate the Earth's radiation budget by interacting with incoming shortwave solar radiation and the outgoing longwave radiation (Schulz et al., 2006; Ghan et al., 2012; Leibensperger et al., 2012; Myhre et al., 2013). However, significant uncertainties still exist in the knowledge on the direct radiative effects of aerosols (Bond et al., 2013; Boucher et al., 2013), both globally and regionally, arising from the limited understanding of the different aerosol types and their spatio-temporal and vertical variations at regional and sub-regional scales (Pöschl, 2005; Remer et al., 2005). A primary source of uncertainty in aerosol-radiation interactions and climate forcing of aerosols is the inadequate representation of the vertical distribution and radiative effects of aerosols, and their spatial variations in climate models (Koffi et al., 2012; Samset et al., 2013; Kipling et al., 2016; Koffi et al., 2016).

Long-term satellite measurements offer a solution to address this issue (Yu et al., 2010; Koffi et al., 2012; Winker et al., 2013; Prijith et al., 2016), despite their inherent uncertainties. Koffi et al. (2016) have reported large discrepancies in the vertical distribution of aerosols between different climate models, particularly over the Indian region, which is attributed partly to the poor availability of ground-based observations to constrain the models. These discrepancies, though are lesser over the oceans (Mann et al., 2014) than the landmass, would lead to significant differences in the aerosol-induced atmospheric heating in climate simulations, as the same aerosol species at different altitudes can lead to different atmospheric heating, owing to the reduction in air density with increasing altitude (Babu et al., 2011). This will also have implications for climate model simulations of Indian summer monsoon (Lau et al., 2006; Bhattacharya et al., 2017).

As such, concerted efforts are being made to improve the accuracy in aerosol characteristics globally (Ramanathan et al., 2001; Anderson et al., 2003; Remer et al., 2008; Vaughan et al., 2009; Yu et al., 2010; Koffi et al., 2012; Yoon et al., 2014; Pozzer et al., 2015). Lidar measurements conducted over the Indian Ocean and a few surrounding locations during the Indian Ocean Experiment (INDOEX) revealed the presence and long-range transport of light-absorbing aerosols into the free atmosphere (Ansmann et al., 2000; Léon et al., 2001; Müller et al., 2001a, b; Franke et al., 2001; 2003). Over the Indian region, several campaigns were conducted using lidars (Satheesh et al., 2006), research aircraft (Satheesh et al., 2008; Babu et al., 2010; Babu et al., 2016), and high-altitude balloons (Babu et al., 2011) to improve the understanding of the vertical distribution of aerosols and their spatio-temporal variations. However, campaign mode observations merely provide glimpses of a larger picture. The Indian sub-continent is home to different types of aerosols that vary spatially and temporally. In an earlier study using spaceborne lidar observations limited to the Indian mainland, the seasonally varying presence of high-altitude (dust dominated) aerosol layers were observed, which leads to a meridional gradient in the vertical distribution of aerosols (Prijith et al., 2016).

The Indian region encompasses the highly polluted Indo-Gangetic Plain (IGP) over the northern side and the arid region (Thar desert) over the western side; both being major sources of distinct aerosol types. The aerosols from these contrasting source regions are advected across the IGP and the north-eastern region (Pathak and Bhuyan, 2014) to the Bay of Bengal (BoB) (Moorthy et al., 2007; Satheesh et al., 2009; Srinivas and Sarin, 2013) and are often observed as elevated layers.





The west to east transport is consistent in the northern parts of India irrespective of the season, though there is a reversal in the synoptic wind direction during post-monsoon and winter in the southern part. Recent aircraft campaigns revealed the presence

of elevated aerosol layers throughout the Indian region in the 1-3 km altitude range (Vaishya et al., 2018; Manoj et al., 2020). Interestingly, the altitudes at which these elevated layers were observed were higher in the north-western part of India compared to the northeast (Manoj et al., 2020). Aerosol concentration within the atmospheric boundary layer (ABL) and free-atmosphere has been reported to exhibit a decreasing and increasing trend respectively over the Indian region (Manoj et al., 2019; Ratnam et al., 2021). This has partly been attributed to an increased convective lifting of aerosols, lofting them from the

surface to the free-atmospheric altitudes. Under such conditions, unless climate models use realistic vertical structures of aerosol distribution, they cannot improve the climate impact assessments over the Indian region (Koffi et al., 2016).

In the light of the above scenario, in this work, we examine the vertical distribution of aerosols, their zonal (east - west) gradients over the Indian mainland and its surrounding ocean and the resulting changes in the aerosol-induced atmospheric heating using long-term multi-satellite observations constrained by in-situ and assimilated observational data sets,

combined with radiative transfer calculations. The observational data, radiative transfer calculations, and the methodology followed in deriving assimilated data sets are discussed in Sect. 2. The vertical structure and zonal gradients of aerosols and associated Particulate Depolarisation Ratio (PDR) are presented in Sect. 3.1 and 3.2 respectively. The zonal gradients in the Single Scattering Albedo (SSA) and the impacts on aerosol-induced atmospheric heating are presented in Sect. 3.3 and 3.4 respectively.

## 2 Observational data and methodology

### 2.1 Spaceborne lidar observations

The observations from the spaceborne lidar – Cloud Aerosol LIdar with Orthogonal Polarization (CALIOP) aboard Cloud Aerosol Lidar and Infrared Pathfinder Satellite Observations (CALIPSO) satellite form the primary database for the present study. CALIOP uses an active laser remote sensing technique and provides measurement data during both day and night. Lidar

profiles collected from 2007 – 2020 have been used to generate a statistically robust mean picture of the vertical distribution of aerosols over the Indian region. Level-2, Version 4.20 (Young et al., 2018; Liu et al., 2019) clear-sky, day and night layer product of aerosol extinction coefficient for 532 nm wavelength has been used to analyze the vertical structure of aerosols within the region 55 – 95° E and 0 – 32° N, encompassing the Indian landmass and the adjoining oceans, as shown in Fig. 1.

In view of the reported large spatial and temporal variations in the aerosol properties (Satheesh et al., 2006; Moorthy

et al., 2009) and meridional gradients in the vertical distribution of aerosols over the Indian region (Prijith et al., 2016), the study region has been divided into three sub-regional latitude bands: (a) Sub-Region 1 (SR1; 22 – 32° N), which covers north India, (b) Sub-Region 2 (SR2; 10 – 22° N), which covers central and peninsular India, Arabian Sea (AS), and the BoB, and (c) Sub-Region 3 (SR3; 0 – 10° N), which covers the southern tip of peninsular India and the northern Indian Ocean, as shown in



Fig. 1. The present study has been carried out during the four seasons: winter (DJF; December – February), pre-monsoon (MAM; March – May), monsoon (JJAS; June – September), and post-monsoon (ON; October – November).

The CALIOP-derived Level 2 aerosol extinction coefficient profiles were subjected to standard screening procedures (Winker et al., 2013). Aerosols were identified using the Atmospheric Volume Description (AVD) flag. Data points having Cloud-Aerosol Discrimination (CAD) score outside -80 to -100 were identified as clouds/non-aerosols and were eschewed from the analysis (Liu et al., 2009). Extinction quality control (QC) flags have been used to account for the error in extinction coefficient retrieval resulting from the lidar ratio adjustments incurred during the data inversion. For example, if the first bit of the QC flag is 0 (1), it refers to the lidar ratio remaining unchanged (constrained) during the lidar retrieval process (Young et al. 2018). CALIOP makes use of different lidar ratios to capture different aerosol species, with the values ranging from 23 sr for clean marine aerosols to 70 sr for polluted continental aerosols at 532 nm wavelength (Kim et al., 2018). Data having uncertainty flag values above 99.99 $km^{-1}$, which indicates that the iterations for the solution did not converge, have been omitted in the final stage. The resulting vertical profiles provide a qualitative but not quantitative representation of the vertical structure of aerosols, due to the uncertainties involved in the lidar ratio (extinction to backscatter ratio) assumption during the retrieval of aerosol extinction coefficient from the lidar backscatter signals. To overcome this limitation, the aerosol extinction coefficient profiles have been normalized using the combined Level-3 Version 6.1 average aerosol optical depth (AOD) from Moderate Resolution Imaging Spectroradiometer (MODIS) on board Aqua and Terra satellites (Wei et al., 2019). The resulting profiles will then capture the realistic vertical distribution (offered by CALIOP) and the columnar loading (offered by MODIS) of aerosols. Dust aerosols have been identified using the PDR profiles from CALIOP. Both Level-2 aerosol extinction coefficient and PDR profiles at 532 nm from CALIOP have a horizontal resolution of 333 m and a vertical resolution of 30 m in the troposphere (Winker et al., 2009). For deriving the quality assured PDR data, a procedure similar to that employed for extinction coefficient profiles has been followed, except that the PDR QC and PDR uncertainty flags have been used instead of extinction QC and uncertainty flags.

## 2.2 Radiative transfer calculations

The CALIOP-derived vertical structure of aerosols has been used to estimate the layer-wise atmospheric heating effects of aerosols and its zonal gradient over the Indian region. Radiative transfer calculations have been carried out using Santa Barbara DISORT Atmospheric Radiative Transfer (SBDART; Ricchiazzi et al., 1998) code for a plane-parallel and vertically inhomogeneous atmosphere. SBDART calculations are carried out using the discrete ordinate method (Chandrasekhar, 1960; Stamnes et al., 1988) for eight radiation streams and considering an atmosphere having a vertical resolution of 0.5 km from 0 to 10 km and a lower resolution thereafter. The calculations were carried out in all the three sub-regions and all the four seasons for every degree longitude from 65.5° E to 90.5° E. The longitudinal outer bounds are set by the availability of the assimilated AOD and SSA data sets, which will be explained in detail in Sect. 2.3.





Vertical structure of atmospheric thermodynamics pertaining to tropical atmospheric conditions were input to SBDART. The spectral surface reflectance values within the visible and near-infra red wavelengths for all the seasons, corresponding to all the grid points were obtained from MODIS (Surface Reflectance product Daily L2G Global 250m, MOD09A1.006 and MYD09A1.006). Accuracy of critical inputs, such as aerosol scattering phase function and SSA, is important in improving the accuracy of the estimated radiative effects. Since we did not have a species-segregated composition

of aerosols over the entire study region, aerosol scattering phase function values were obtained from a Mie scattering model – Optical Properties of Aerosols and Clouds (OPAC) (Hess et al., 1998). SBDART calculations were made at every 5° solar zenith angle intervals and clear sky conditions for two cases: "without aerosols" and "with aerosols" over the shortwave radiation spectrum. The difference in the layer-wise net radiative forcing between these two cases provide the aerosol radiative forcing (ΔF), which has been used to estimate the aerosol-induced atmospheric heating rate (dT/dt) as given below (Liou,

135     2002).

$$\frac{dT}{dt} = \frac{g}{C_p}\frac{\Delta F}{\Delta P} \qquad (1)$$

where g is the acceleration due to gravity, $C_p$ is the specific heat capacity of air under constant pressure, and $\Delta P$ is the pressure difference between the top and bottom levels in a layer. Further details on the estimation of $\Delta F$ and dT/dt may be found

elsewhere (Babu et al., 2002; Satheesh, 2002; Vinoj et al., 2004; Satheesh et al., 2008) and are not repeated here.

        The radiative forcing of aerosols and the resulting atmospheric heating effects under clear sky conditions are primarily dependent on the aerosol concentration and SSA within each layer. While the CALIOP profiles provide the former data set, a realistic data set of the latter is not readily available. Even though the SSA measurements provided by satellite sensors such as Ozone Monitoring Instrument (OMI) aboard Aura satellite captures the spatial variations in aerosol absorption to a large extent,

their magnitudes have been reported to include large uncertainties, especially over the Indian region (Eswaran et al., 2019), partly attributed to the assumption of aerosol layer heights (Satheesh et al., 2009; Satheesh et al., 2010). A few ground-based measurements, though provide more accurate estimates of columnar SSA, are limited by the sparsity of measurements, and provide rather point values. An optimal combination of the advantages offered by OMI (the accurate spatial variations) and the sparse yet more realistic columnar measurements from in-situ observations are utilized to derive more realistic estimates

of aerosol radiative forcing and atmospheric heating rate. For this, statistical data assimilation techniques have been employed (Kalnay, 2003; Lewis et al., 2006; Pathak et al., 2019). In the present study, an assimilated data set of the aerosol SSA derived by combining OMI and in-situ observational data have been used, as detailed in the next section.

## 2.3 Assimilated SSA data set

Assimilated gridded datasets of SSA covering the entire Indian landmass has been generated for the first time by Pathak et al.

(2019) using long-term ground- and satellite- based measurements of AOD and absorption AOD (AAOD). This dataset, which provides SSA over the Indian mainland at 1°×1° spatial resolution for the three seasons DJF, MAM, and ON, is used in this



study to constrain the absorption potential of aerosols in the radiative transfer calculations. However, as the assimilated datasets are available only over the landmass, we have extended these over to the adjoining oceanic region based on some important observations and assumptions which are explained in detail later in this section. Nair et al. (2008b) observed high concentrations of Black Carbon (BC) away from the coast over the AS and BoB. Consequently, very low values of SSA (~0.85) exists over the BoB away from the coast (Nair et al., 2008a). From Fig. 2, it can be seen that OMI fails to reproduce the low values of SSA over both land (Goto et al., 2011; Srivastava et al., 2011) and ocean (Nair et al., 2008a). Taking this into account, we extended the Assimilated SSA (ASSA) over to the ocean instead of directly using the OMI SSA (OSSA) for radiative transfer calculations.

For this, we started by choosing two regions consisting of both land and ocean, one including the AS (west) and the other including the BoB (east). These regions are represented by the dashed rectangles in Fig. 1. The mean values of OSSA in these regions for the different seasons are given in Table 1. The table shows the mean SSA over land and ocean separately. It can be seen that OSSA shows weak seasonal variations over both land and ocean. The seasonal changes are similar over land and ocean with highest values during ON and lowest values during MAM. In general, the magnitudes of OSSA over the land are marginally lower. In the west, during DJF and MAM, the SSA over ocean is marginally higher. Next, we estimated the difference between OSSA and ASSA for these regions considering only the data points over the land. In our study, the ASSA values are considered over the land region and to overcome the OMI overestimation over the ocean, we extended the ASSA to the oceanic regions as well. For this, we made the following assumptions: (1) OSSA and ASSA values at the coast should be the same, (2) As we move away from the coast, the difference between OSSA and ASSA vary as a function of the longitude.

The extension was done separately for the regions on the east and west considering one latitude at a time. We started the extension by estimating the meridional average of the OSSA over the ocean (AS and BoB, represented by the dashed rectangles in Fig. 1). Linear regression is performed on the mean OSSA values and these regression coefficients were used to estimate the SSA at the coast corresponding to individual latitudes. The ASSA over the land has sharp spatial variations and hence the mean SSA for a 2°x2° grid close to the coast was estimated. The difference between this mean SSA value and the value obtained for the coast using the regression coefficients was estimated and is referred to as δSSA hereafter. Since the effect of anthropogenic sources reduce as we move away from the coast, the zonal variation in δSSA was parameterised as a function of longitude ($\delta SSA_{lon}$). Removing $\delta SSA_{lon}$ from the OSSA, the extended SSA (eSSA) for a particular latitude is estimated. This is repeated for each latitude and the eSSA for the whole oceanic region is estimated. After completing the extension, we have eSSA values for oceanic region above 8°N. Since we do not have ASSA to constrain the model below this latitude, the same method cannot be employed. To overcome this limitation, we estimated the difference in OSSA and eSSA over the oceanic region (Table 2). Assuming a constant bias in the OSSA in this region, the eSSA values are estimated by subtracting the bias from the OSSA values. This method of extension ensures that the spatial and zonal variations in OSSA are retained. Finally, merging eSSA over ocean and ASSA over land, we obtained the combined SSA (cSSA) for the study region. The OSSA and the cSSA for the three seasons DJF, MAM, and ON are shown in Fig. 2. The method employed in this study retains the spatial variations in OSSA and offers the realistic lower values and sharp changes near the coast. The eSSA



values have been compared with the AErosol RObotic NETwork (AERONET) retrieved SSA values from Maldives Climate Observatory at Hanimaadhoo (MCOH), a remote, island station located in the Indian Ocean (Table 3). The values are comparable during DJF and MAM while eSSA values are marginally higher during ON. The SSA from extensive measurements over the entire BoB during Integrated Campaign for Aerosols, gases, and Radiation Budget (ICARB) – 2006
(March-May) revealed values in the range 0.89 - 0.96 (Nair et al., 2008a), which is in line with the cSSA estimates in the present work. This cSSA has been used in the radiative transfer calculations explained in Sect. 2.2.

## 3 Results and discussions

### 3.1 Aerosol Extinction Coefficient

The zonal variations in the seasonally averaged vertical distribution of aerosol extinction coefficients over the study region are
shown in Fig. 3. The three sub-regions are shown in the panels from top to bottom (SR1 to SR3) and in each row, the seasonal changes are shown from DJF (left extreme) to ON (right extreme) through MAM and JJAS seasons. Significant intrusion of aerosols into higher altitudes is seen; comparatively more in the north (SR1) and least in the south (SR3), irrespective of the seasons. In all the three sub-regions, the vertical extent is minimum in DJF and is maximum in MAM and JJAS. The vertical transport of aerosols to altitudes above 6 km is controlled by dynamics (Ratnam et al., 2018). Aerosols are mostly limited to
below 3 km in DJF while they are present even above 5 km during MAM and JJAS. The maximum vertical extent of aerosols during JJAS is above 6 km for SR1, around 5.5 km for SR2, and within 4 km for SR3. This is in line with earlier reports from campaign measurements using airborne lidars (for e.g., Satheesh et al., 2008) as well as using satellite data (Prijith et al., 2016), where meridional (north to south) gradients have been reported in the vertical distribution of aerosols over the Indian mainland. The difference in the vertical extent of aerosols observed in Fig. 3 for all the sub-regions and seasons is due to the existence
of this north-to-south gradient. However, the zonal (west-to-east) gradients remained unexplored until this study.

Figure 3 also reveals that the extent of vertical intrusion has an eastward gradient, as revealed by the profiles of the blue shades in most of the panels; being highest in west and sloping downwards towards east, as indicated by the dotted yellow line in Fig. 3c as illustrative. In general, the magnitudes of the extinction coefficients are higher close to the surface and decreasing with altitude. In SR1, such a feature is observed to be the strongest within the longitude region 70 – 80° E whereas
in SR2, it is more widespread. The vertical extent is controlled by the combined actions of the convective lofting and the consequent dynamics of the ABL, long range transport, scavenging by clouds and precipitation and the proximity to the sources. ABL is shallower during DJF and ON over the Indian region due to the lower surface temperatures and the resulting reduced convective mixing processes (Sathyanadh et al., 2017), particularly so in SR1, where the winter temperatures come down to a few °C to near zero. As a result, during these seasons, the vertical extent of aerosols remains low due to lack of
mixing. On the other hand, during MAM and JJAS, when the convection is stronger (temperature goes well above 40° C in



SR1) vigorous convective lifting prevails over the Indian region. In addition to this, effects of long-range transport are also observed in the northwest (<70°E; in SR3 and SR2) during this period.

Closer examination of the seasonality in the longitudinal variations in Fig. 3 reveals that:

1) During DJF (Fig. 3 a, e, i), the vertical extent of aerosol extinction peaks within 75 – 80° E and drops off as we move further to the east or west. The peak in the vertical distribution is observed to be above the surface (within 0.5 – 2 km) in SR2 as compared to the surrounding oceanic regions. This general pattern is weakest in SR1 (mostly land) and clearly visible in SR2 (mixture of land and ocean). In SR1, the near-surface aerosol extinction coefficient values are highest in the 70 – 90° longitude band (Fig. 3a). This can be attributed to the confinement of aerosols near the surface by the shallow winter ABL and the weak winds typical to IGP during that season, leading to low ventilation coefficients, as has been reported earlier in campaign studies (Nair et al., 2007). In the 55 – 70° longitude band, the magnitudes of near-surface aerosol extinction coefficients are lower as this region consists of the ocean in SR2 and SR3. Nevertheless, the vertical extent of aerosols remains comparable to that in the 70 – 90° longitude band. In SR2 (Fig. 3e), the general pattern is very similar and the fall in the vertical extent is sharper towards the west as compared to the east. It is worth noticing that in the 85 – 95° longitude band, composed mostly of BoB, the near-surface values of extinction coefficients are comparable or even higher than that over the land. This feature has been reported in several earlier studies as well and are attributed to the outflow of aerosols from the IGP to the BoB (Banerjee et al., 2019; Kompalli et al., 2021). In SR2, the land-sea boundaries on the west (east) are around 72.5°E (85°E) longitude, which changes to 75°E (80°E) in SR3, reducing the width of the land region. The general pattern described above, i.e., the vertical extent being higher over the land region, is observed to be modulated by this meridional change in the land-sea border. Both the width of the central peak and its vertical extent reduces from SR2 to SR3 (see for e.g., Fig. 3 e, i). During MAM and JJAS, when the transport of mineral dust is strong, the zonal variations exhibit a slightly different pattern with considerable amount of aerosols above 2 km. The west to east decreasing gradient in the vertical extent of aerosols is clear during this period. It can also be seen that as the influence of dust subsides, the vertical extent decreases and reaches a minimum by DJF.

2) As the season advances to MAM, even though the zonal gradient remains weak, the vertical extent of aerosol extinction increases, and their near-surface values decrease (compared to DJF), which is attributed to the enhanced convective mixing and the consequent lofting of aerosols over the land.

3) A negative zonal gradient emerges out clearly during JJAS with the vertical extent increasing in the west and decreasing in the east. Earlier studies have reported the transport of dust aerosols from west Asia and east Africa during this period across the IGP (Prijith et al., 2016; Banerjee et al., 2019). The emergence of a negative zonal gradient in SR3, and the increase in vertical extent over the western region in particular, during JJAS could be attributed to this. A similar negative zonal gradient is sustained during ON but with a reduced vertical extent. The transport from the west weakens and its impacts on the aerosol loading becomes negligible and the conditions during





ON closely resemble DJF. A striking feature seen in SR3 is the occurrence of an elevated aerosol layer with high extinction coefficient between 2 – 4 km altitudes during JJAS.

4) SR2, which has land region sandwiched between the AS and the BoB, provides a clear signature of the vertical structure of aerosols; the prominent feature being the higher vertical extent over the land with the extinction coefficient maxima occurring within 1 – 2 km above ground, in general conformity with earlier reports (Moorthy et al., 2004; Satheesh et al., 2006; Prijith et al., 2016; Prasad et al., 2019). During DJF, the eastern side exhibits higher concentration of aerosols (below 1 km) compared to the western side which may be due to the larger share of land in the eastern side compared to the western side. Consequently, the ABL dynamics may exhibit larger temporal variations on the eastern side leading to larger confinement of aerosols close to the surface during DJF. During MAM, the vertical extent increases throughout SR2, but interestingly, the altitude above which the extinction coefficient is less than 0.1 km$^{-1}$ does not exhibit a similar pattern and is observed to be the least around 65°E, where the land-ocean boundary of the Indian peninsula commences in SR2. This length scale marks the influence of near-surface aerosols in the vertical distribution and raises another question of why the vertical extent does not follow such a pattern. It should be noted that during MAM, the dust transport from the Arabian Peninsula might be the reason why the aerosol concentration is not close to zero at higher altitudes. This argument becomes clear during the transition to JJAS when the dust transport is strong and leads to an increased vertical extent and negative zonal gradient in SR2. ON marks the return to the vertical structure and its zonal variations similar to that observed in DJF.

5) A plume like structure, observed in the 55 – 70°E longitudes in SR1, is small in size and at lower altitudes in DJF, but becomes larger and extents to ~2.5 km in MAM. But, compared to SR1, the plume in the western region is absent in SR2 during DJF and MAM but appears with a larger spatial extent in JJAS, more or less appearing as a high-altitude layer of aerosols all the way from 55°E to 65°E. This leads to a dual peak vertical structure in JJAS in SR2. Interestingly, the eastern side of the plume overlaps roughly with the western boundary of the peninsula.

6) SR3 is mostly oceanic with very less share of land; consequently, the zonal gradients are weak in all the seasons except JJAS and exhibits a higher vertical extent around 75 – 80°E where land is present.

7) During JJAS, all the regions exhibit a negative zonal gradient of aerosol extinction coefficient due to the strong inflow of aerosols brought about by the south-westerly winds. The west to east gradient is strongly modified by the west Asian (including the Thar Desert and the semi-arid regions of north-western India) dust transport across the IGP and over to the BoB (Prasad and Singh, 2007; Kumar et al., 2008). These regions, including the contiguous arid regions of West Asia, are known to be strong sources of mineral dust, which are advected across the IGP aided by the favourable meteorological conditions (Moorthy et al., 2005; Niranjan et al., 2007; Beegum et al., 2008; Vaishya et al., 2018; Manoj et al., 2020). The anthropogenic dust source can contribute ~45% to the total dust contribution over the Indian region and the one major source region is IGP (Ginoux et al., 2012). Previous studies using ground-based and spaceborne lidars have shown elevated aerosol layers, reaching altitudes as high as 5 km, over IGP during MAM and JJAS (Gautam et al., 2010; Mishra and Shibata, 2012). Furthermore, the vertical extent of aerosol extinction



coefficient increases around 55° E. This is due to the portion of the Middle East, a major dust source to the AS (Namdari et al., 2018), included in the north-west part of SR2. The dust transport has been reported to cause a
strengthening of the Indian summer monsoon through tropospheric heating (Jin et al., 2014; Vinoj et al., 2014; Solmon et al., 2015). These observations suggest that the zonal gradients in the vertical extent of aerosols is largely controlled by the long-range transport of dust. With a view to examining whether this elevated aerosol layer in JJAS is formed by advected dust aerosols, we looked into the PDR profiles from CALIOP.

### 3.2 Particulate Depolarisation Ratio

The seasonality in the zonal variations of the PDR profiles over the three sub-regions is shown in Fig. 4. It emerges that SR1 is the region most impacted by the dust transport throughout the year. The non-dust and fine dust aerosols are identified by the PDR range 0.02 – 0.07 and 0.2 – 0.4 respectively (Gautam et al., 2009; Groß et al., 2011; Lakshmi et al., 2017). Long range transport of dust from the west across the northern AS towards the Indian mainland has strong influence on the aerosol loading and composition over the Indian region (Moorthy and Saha, 2000; Li and Ramanathan, 2002; Prijith et al., 2012; Banerjee et
al., 2019). We observe that the PDR values are mostly in the range 0.2 – 0.3 in SR1 (Fig. 4a – d). High PDR values between 0.2 – 0.4 are prominently seen over the western arid parts associated with the dust sources. However, these high values are limited to longitudes below 80°, except during MAM in SR1 and SR2, and during JJAS in SR3.

The influence of mineral dust is evident throughout the year over SR1; highest in MAM and lowest in DJF. Vaishya et al. (2018) have shown that the dust fraction was ~0.5 in the west and central IGP during pre-monsoon while it was negligibly
small in the eastern IGP. A similar seasonal variation is observed over SR2 where PDR values above 0.3 are limited to the east of 70°E except in JJAS. In SR3, the influence of dust is much lower and PDR values above 0.2 are observed only during JJAS. However, the extinction coefficient values are less than 0.1 km$^{-1}$ above 4 km in this region indicating that although PDR values are high, the impact will be negligible due to the low aerosol concentrations (Fig. 4k).

During MAM and JJAS, high PDR values are observed throughout the column extending from close to the surface to
above 6 km over the west (mostly at longitudes >70°E). However, the extinction coefficient values are below 0.05 km$^{-1}$ above 5 km. Modelling studies (Banerjee et al., 2019) have delineated the different pathways of dust transport from local and remote sources across IGP and the distinctiveness in their vertical distribution. The strength of dust transport leads to the scattering aerosol concentration observed over south-eastern part of AS and northern Indian Ocean to be comparable to that observed over the western coast of India (Kompalli et al., 2021). With the exception of the northern BoB which is affected by local
sources, BoB is affected by the remote sources all through the year and this intensifies during monsoon (Banerjee et al., 2019; Kompalli et al., 2021). The elevated dust traveling across the IGP becomes more absorbing when fine BC (produced by anthropogenic activities in the IGP) gets coated/adsorbed on the porous dust (Moorthy et al., 2007; Prasad and Singh, 2007; Pandithurai et al., 2008; Brooks et al., 2019). The higher hematite content in the soil also adds to the enhanced dust absorption (Deepshikha et al., 2005; Moorthy et al., 2007). It was shown that these coated aerosols have east to west contrast with higher



coating of BC in the eastern part of the northern Indian Ocean (outflow to BoB through IGP) compared to the western part (outflow from west coast of India; Kompalli et al., 2021). The radiative impact of these aerosols should be evaluated taking into consideration the extinction coefficients and the absorbing nature of the aerosols. The aerosol absorption over the region has been examined using cSSA explained in Sect. 2.3 and is discussed below.

### 3.3 Combined SSA

SSA is an important parameter controlling the radiative effects of aerosols. Ideally, it is desirable to have altitude-resolved measurements of SSA as it can reproduce more realistic estimates of atmospheric heating induced by aerosol layers at different altitudes. However, the availability of SSA from in-situ measurements is limited over the Indian region and satellite measurements (OMI) tend to overestimate the regional SSA. Estimates of aerosol radiative effects are often made by relying on the column average values of SSA (Dubovik et al., 2000; Satheesh et al., 2010; Choi and Chung, 2014), which can lead to

uncertainty in the vertical profiles of aerosol radiative forcing and the resulting atmospheric heating effects. It can be observed from Fig. 2 that cSSA provides more realistic values of SSA over the oceanic region.

        A comparison of the zonal variations in the cSSA and OSSA corresponding to the different seasons during which ASSA is available, is shown in Fig. 5. cSSA shows stronger absorption and significant zonal variation in most parts of the sub-regions compared to the OSSA. The reported low SSA over the IGP region (Di Girolamo et al., 2004; Ramana et al., 2004) is

well captured by cSSA along with the high SSA over the western part of India during MAM and JJAS due to the inflow of dust (Badarinath et al., 2009; Das et al., 2020). The low SSA during MAM and high during DJF are also observed in cSSA. As expected, SR3 exhibits the highest SSA, while SR1 has the lowest values, well in accordance with the fact that SR3 has limited sources of absorbing aerosols compared to SR1. While OSSA captures the spatial variations quite well, the absolute magnitudes are high and unrealistic due to reasons explained before. cSSA values are observed to sort this out and provide

more realistic values in addition to the accurate spatial variations observed in OSSA. The seasonal low over SR1 is observed during MAM with an average of 0.87 ±0.02, while for DJF and ON, the average values are 0.89 ±0.02 and 0.88 ±0.02, respectively. The zonal variations of cSSA in SR1 are similar during DJF and ON, but there is a visible drop in the magnitudes near 90°E in ON.

        SR1 consists mostly of land areas of the northern part of India, over which advected dust and local anthropogenic

emissions contribute significantly (Kedia et al., 2014), leading to the lowest SSA over SR1 (0.89 ±0.02 in DJF, 0.86 ±0.02 in MAM, and 0.88 ±0.02 in ON). This sub-region has densely distributed coal-fired thermal power plants and several small and large-scale industries (including cement factories and iron and steel mills), the emissions from which contribute significantly to the aerosol loading throughout the year (Garg et al., 2001; Reddy and Venkataraman, 2002; Prasad et al., 2006; Gadi et al., 2011). Low SSA values over this region are consistent with the previously reported in-situ measurements (Verma et al., 2013;

Babu et al., 2016; Vaishya et al., 2018) as well as the high concentrations of absorbing BC (for e.g., Prasad et al., 2006; Badarinath et al., 2009; Eck et al., 2010; Giles et al., 2011).





During MAM, the SSA values decrease in the western part of India (Fig. 5b) and this increase in SSA coincides with the influx of dust (Fig. 4b). A corresponding change is observed in the vertical extent of the aerosol extinction coefficient in the west (Fig. 3 b, f, j). The PDR values in the range 0.2 – 0.4 indicates strong influence of dust during MAM (Fig. 4 b, f). The effect of dust over the Indian region starts during MAM, peaks during JJAS, and drastically reduces during ON and reaches a minimum during DJF (Banerjee et al., 2019).

In general, the SSA values in SR2 (0.92±0.01 in DJF, 0.91±0.01 in MAM, and 0.93±0.01 in ON) are higher than those in SR1, but the low values of SSA in the longitude band 75-85°E in SR2 clearly indicate the influence of highly absorbing anthropogenic emissions over the land. The aerosols in the west are clearly much less absorbing as compared to those in the IGP outflow region (near 85°E). Thus, there is a west-to-east decrease in SSA. Beyond 85°E, SSA starts increasing due to the relatively cleaner nature of the oceanic region. During DJF, SSA is low in the 67-72° longitude band but as dust intrudes this region in MAM, SSA sharply increases, as dust is less absorbing than BC. In SR3, the sub-region with highest SSA, the pattern of zonal variations (0.95±0.03 in DJF, 0.96±0.01 in MAM, and 0.97±0.01 in ON) closely resembles that in SR2.

### 3.4 Zonal variations in aerosol-induced atmospheric heating

The information on the height-resolved aerosol extinction coefficients and SSA (cSSA), and their spatial and seasonal variations have been used to estimate 3-D aerosol radiative heating, for the first time over the Indian region, following the methodology detailed in Sect. 2.2. The zonal variations in seasonally averaged dT/dt profiles over the three sub-regions are shown in Fig. 6. In general, dT/dt is high near the surface and decreases with altitude throughout the study region, closely following the vertical distribution of aerosols. The magnitude of dT/dt increases from south to north, i.e., they are highest in SR1 and lowest in SR3 irrespective of the season.

During MAM, the effect of elevated aerosol layers on dT/dt profiles is clearly discernible between 2 km and 4 km within the longitude range 70 – 90°E (Fig. 6 b, e, h). The magnitude of dT/dt in these layers is ~1 K d$^{-1}$ and is higher compared to the values in adjacent altitude levels. Such layers, observed in SR2, extend up to ~3 km with a peak near 2.5 km (~0.8 K d$^{-1}$). In SR3, the vertical extent and magnitude of dT/dt are much lesser within 1 km altitude, and the higher values beyond 1 km are observed close to the land between 75 – 80°N. It has been observed that the dust optical depth over the BoB significantly increases during MAM in the 10-20°N latitude band (Lakshmi et al., 2017).

Higher values of dT/dt are observed more on the western side in SR1 and SR2 during ON (Fig. 6, f). In the western side (~70°E), these larger values of dT/dt extend to relatively higher altitudes during ON as compared to DJF (Fig. 6 a, d). Due to the weak precipitation in the western region, the influence of dust is still observed in this region, which is also reflected in the high PDR values in the region (Fig. 4d). It may be noted that the zonal variations in the heating rate profiles do not exactly follow the pattern observed in the extinction coefficient profiles which is due to the differences in the aerosol types across the study region. Srivastava et al. (2012) showed that compared to the pristine dust in western India, the dust over central IGP is more absorbing due to the mixing with anthropogenic pollutants. Vaishya et al. (2018) showed that altitude resolved in-situ





SSA measurements across the IGP was lowest over the central IGP and highest in the western IGP during MAM. However,
while using column integrated values of SSA, the heating rates will be underestimated or overestimated depending on whether
the SSA at the particular altitude level is overestimated or underestimated (Manoj et al., 2020).

Zonal variation of the columnar mean dT/dt for the different regions were estimated for the period when ASSA is available (DJF, MAM, and ON), and the results are shown in Fig. 7. During DJF, the mean values are 0.23, 0.13, and 0.05 K d$^{-1}$ for sub-regions SR1, SR2, and SR3, respectively. During MAM, these values attain their annual high of 0.32, 0.17, and 0.05 K d$^{-1}$ which decrease during ON to 0.27, 0.13, and 0.04 K d$^{-1}$ respectively in SR1, SR2, and SR3. It is interesting to note that the seasonal variation is significant in SR1 with the highest values in MAM and the lowest values in DJF. The magnitude of dT/dt in SR1 is almost the double of that observed in SR2. On the other hand, SR3 has the least magnitudes and the weakest seasonal variations with the values remaining around ~0.05 K d$^{-1}$ during all the seasons. It was discussed in Sect. 3.1 that the maximum vertical extent of aerosols is observed around 75°E and decreasing to both sides; a similar pattern can be observed in the case of columnar mean dT/dt as well, as shown in Fig. 7. To quantify this, we selected the longitude bands 65.5 – 77.5 and 77.5 – 90.5 and linear regression fits were made between the columnar mean dT/dt and longitude. The regression coefficients are shown in Table 4; it can be seen that the slopes of the linear regression fits are positive in the 65.5 – 77.5 and negative in the 77.5 – 90.5 longitude bands respectively, as expected. This variation is consistent over the Indian region.

The nature of the spatial gradients observed in this study, with landmass in between AS and BoB, is consistent with that over the oceanic regions reported by Nair et al. (2013). Our results show that the pattern is consistent irrespective of seasons. The heating rates peak around the middle of the Indian landmass and decreases on either side. The anthropogenic pollutants, especially the BC aerosols, are abundant over the landmass, and as the distance from the landmass increases, the influence of the pollutants decreases. The observed high-altitude atmospheric heating imparted by aerosols can increase the atmospheric stability, suppress the convective lifting, and possibly influence the synoptic circulation and precipitation patterns over the Indian region. Regional weather modelling studies making use of realistic aerosol observations can shed more light into the role of individual species in bringing about the observed changes.

**4 Conclusions**

The vertical structure of aerosols, the atmospheric heating imparted by them, and the seasonality of its spatial (zonal) gradients across the Indian landmass and the adjoining oceans have been characterized for the first time using long-term (2007 – 2020) data of altitude-resolved aerosol extinction coefficients. An assimilated SSA data set, derived from in-situ and satellite observations, has been used to obtain realistic aerosol radiative heating estimates. The important findings are given below:

1) Strong seasonal variations are observed in the aerosol extinction coefficient profiles over the Indian region. Two distinct patterns are observed one with a higher vertical extent over the western part of India and the other in which the vertical extent increases from west to east, attains a maximum in the centre, and decreases thereafter. The first



pattern prevails over the northern Indian plains throughout the year and over the central and southern peninsula only during summer/monsoon seasons.

2) The magnitudes of extinction coefficients also show strong altitudinal variations with the surface concentrations constantly decreasing with increasing altitudes in the southern peninsula. On the other hand, over central India and northern plains, it increases initially with altitude to reach a peak and decreases thereafter.

3) The western region, which acts as a gateway for the transported dust reaching the Indian region, is marked by an abundance of dust aerosols. The dust transport modulates the existing zonal gradients and its influence decreases towards the east. The vertical extent of dust aerosols is highest during JJAS over northern and central India.

4) The aerosol radiative forcing and the subsequent atmospheric heating rates exhibit strong zonal variations. In northern India, there is a weak positive gradient in DJF which reverses during ON. In the central and peninsular India, AS,

BoB, and the northern Indian Ocean, the zonal variations and vertical extent are identical during DJF and ON. During MAM, elevated heating layers are observed throughout the Indian landmass around 2 km altitude.

## Data availability

Processed data are available upon request.

## Author contribution

NKK, SKS and KKM together conceived the work. NKK carried out the scientific data analysis and along with MRM, NSA, KKM and SKS were involved in the scientific interpretation of the results, leading to the formulation of the manuscript. HSP provided the assimilated SSA. NKK prepared the initial draft with inputs from MRM and NSA. All authors reviewed the manuscript.

## Acknowledgements

CALIPSO data used in this study was obtained from the NASA Langley Research Center Atmospheric Science Data Center. MODIS level 2 surface reflectance was acquired from https://ladsweb.nascom.nasa.gov/. We acknowledge the mission scientists of MODIS (for level 3 AOD and level 2 surface reflectance) and OMI (for the SSA) and the associated NASA personnel for the data used in this work. We thank AERONET for providing SSA data through their website https://aeronet.gsfc.nasa.gov/. One of the authors (SKS) would like to thank SERB-DST for the J. C. Bose Fellowship and

Tata Education and Development Trust for the support. We thank Divecha Centre for Climate Change for supporting this work.



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





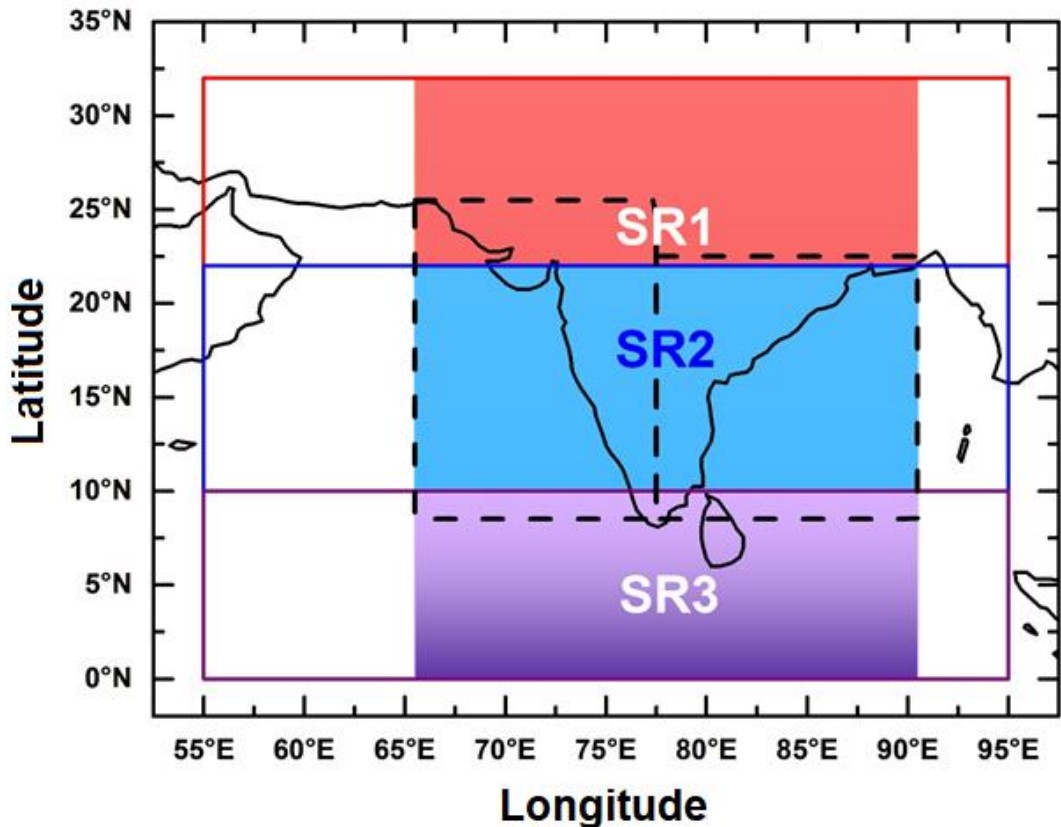


**Figure 1: The study region is divided into three sub-regions. All the sub-regions cover the longitude range from 55°E to 95°E. The latitude bands covered by the sub-regions SR1 (Red), SR2 (Blue) and SR3 (Purple) are respectively 22 - 32°N, 10 - 22°N and 0 - 10°N. The sub-regions are marked by the solid rectangular boxes. The aerosol SSA values have been estimated within the latitude band 65.5 - 90.5°E represented by the filled rectangular boxes within the sub-regions. The dashed rectangular boxes represent the**

**regions selected for extending the assimilated SSA to the oceanic region. The dashed rectangular boxes on the western and eastern sides of peninsular India represent 8.5-25.5°N and 65.5-77.5°E and 8.5-22.5°N and 77.5-90.5°E regions respectively.**





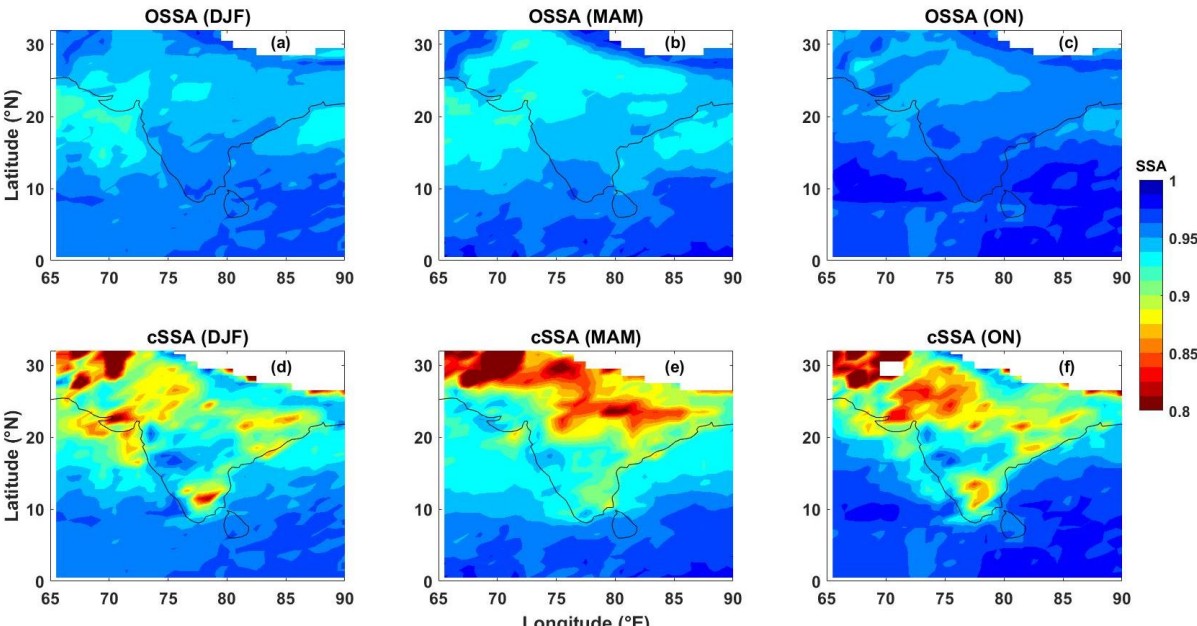

Figure 2: Spatial map of OSSA (panels a, b, and c) and cSSA (panels d, e, and f). The left, middle, and right panels represent the seasons DJF, MAM, and ON respectively.



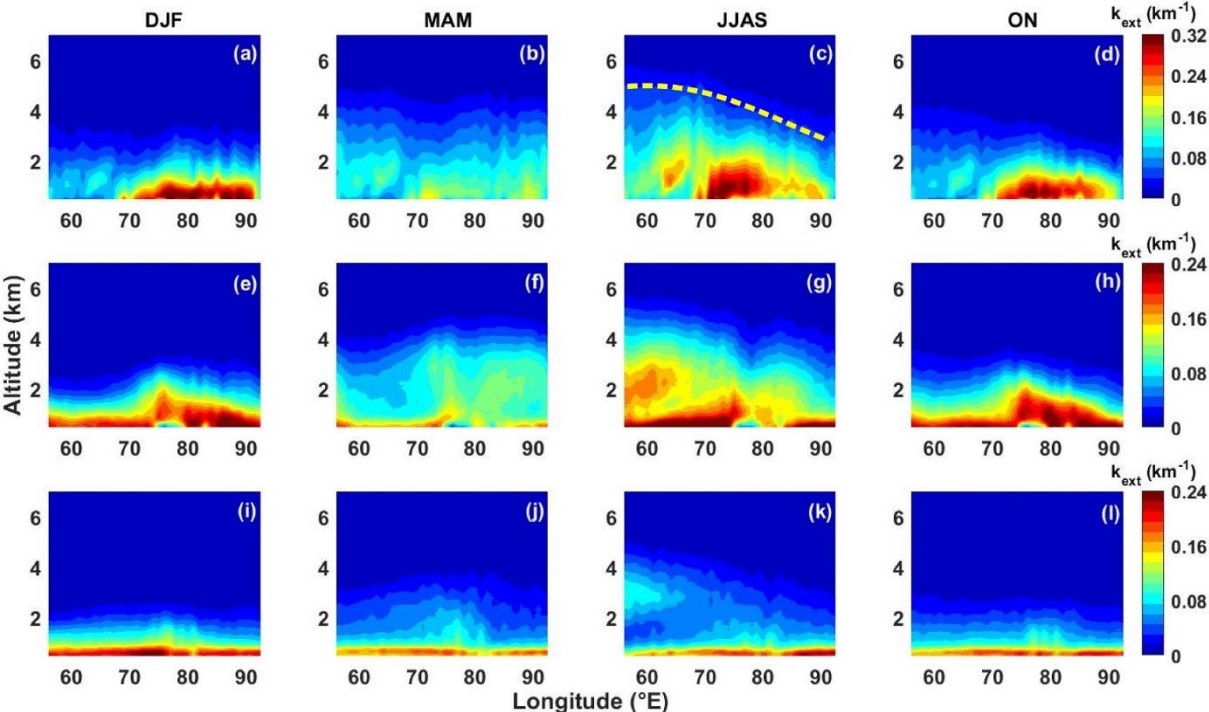

**Figure 3: Zonal variation of the aerosol extinction coefficient (kext) profiles for SR1 (top panel), SR2 (middle panel), and SR3 (bottom panel) sub-regions. Each column corresponds to a particular season, as marked above them.**






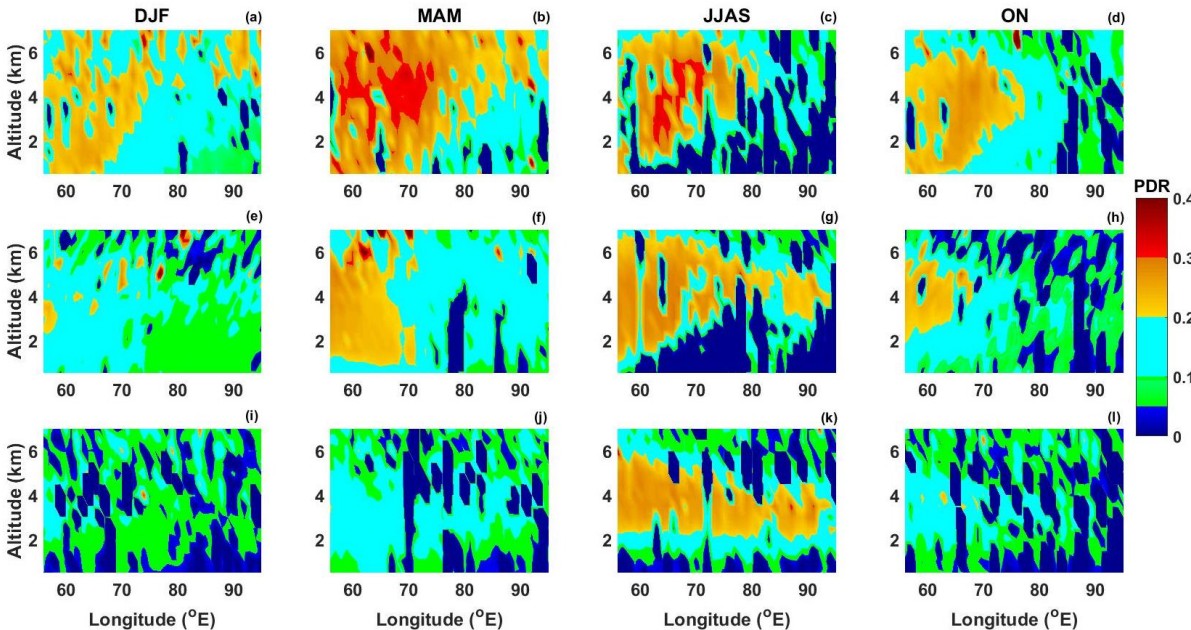

**Figure 4: Zonal variation of PDR profiles for SR1 (top panel), SR2 (middle panel), and SR3 (bottom panel) sub-regions. Each column corresponds to a particular season, as marked above them.**





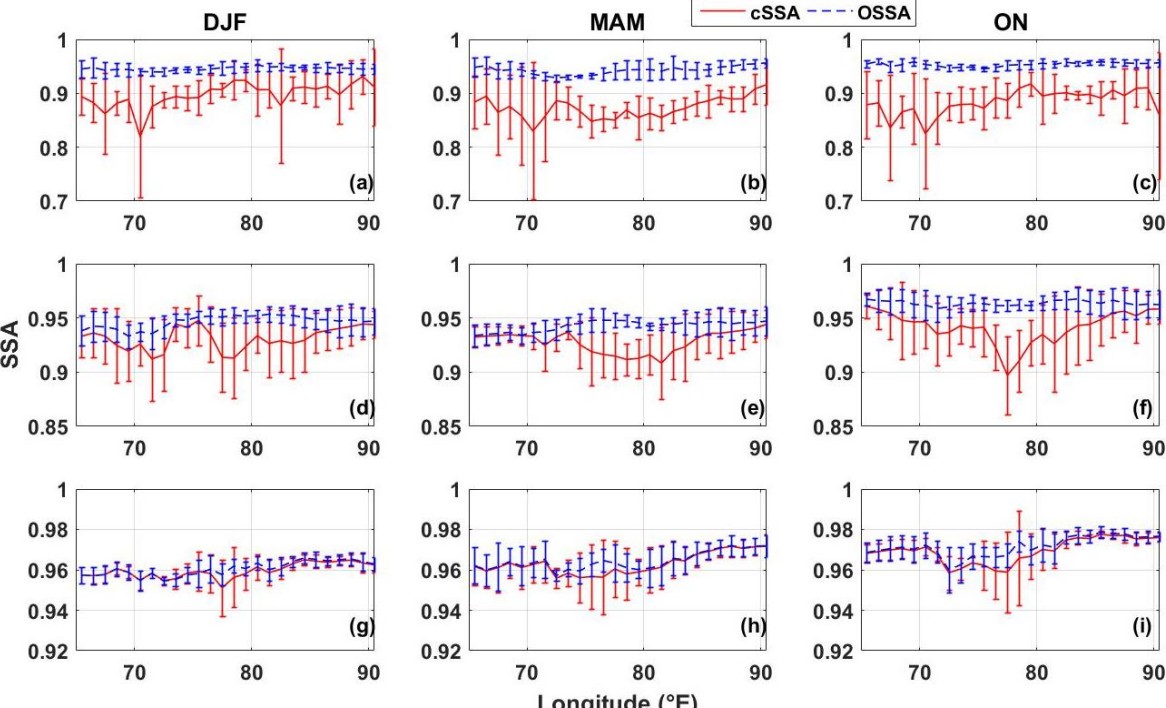

**Figure 5: Zonal variation of cSSA and OSSA in SR1, SR2, and SR3 during DJF, MAM, and ON seasons. The rows from top to bottom correspond to sub-regions SR1, SR2, and SR3 respectively.**





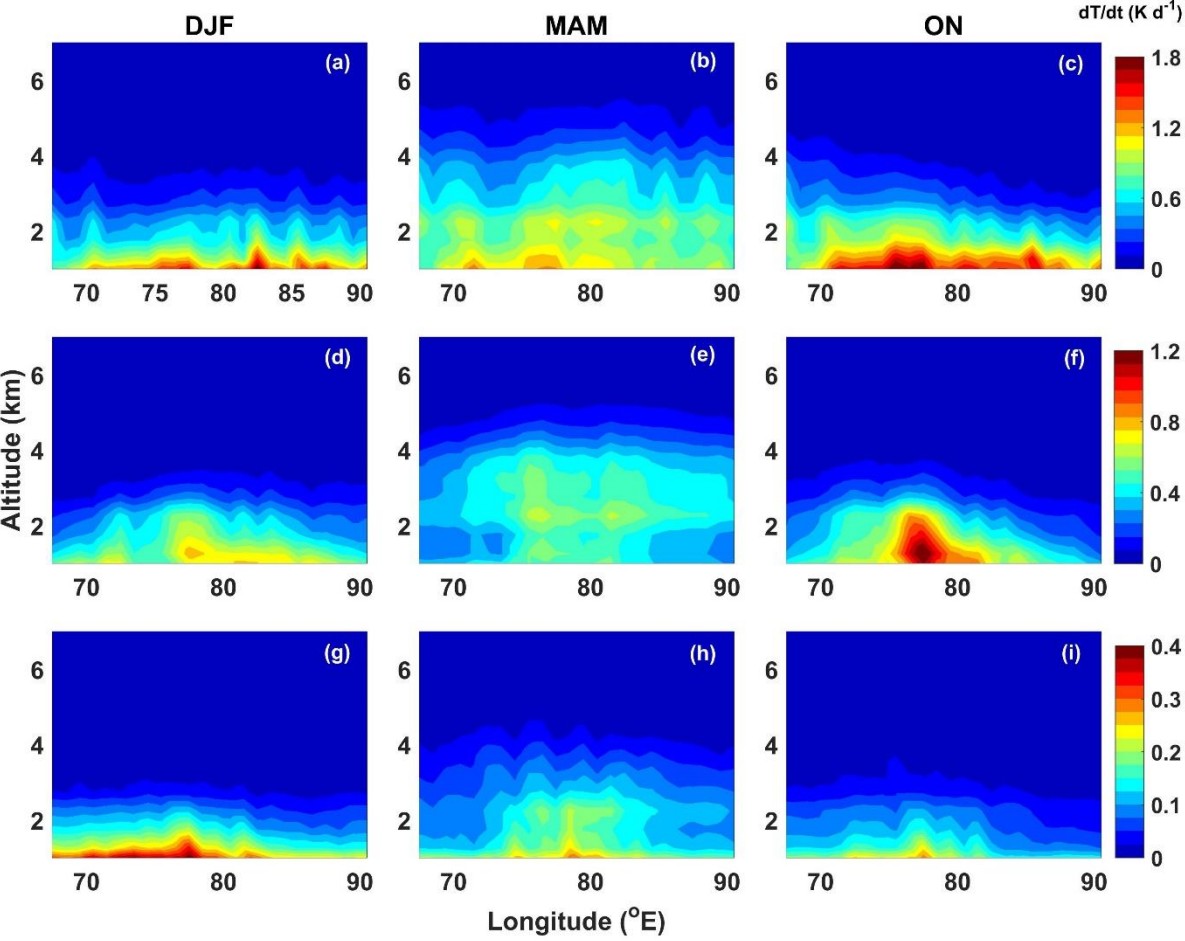

Figure 6: Zonal variation of aerosol-induced atmospheric heating rate profiles (dT/dt) for SR1 (top panel), SR2 (middle panel), and
SR3 (bottom panel) sub-regions. Each column corresponds to a particular season, as marked above them.



**Figure 7: Zonal variation of columnar atmospheric heating rates (dT/dt) for (a) DJF, (b) MAM, and (c) ON seasons. The error bars represent the standard error.**





**Table 1: Mean OSSA corresponding to the two dashed rectangles (on the west and east of the Indian mainland) shown in** Error! Reference source not found.**. The suffixes L and O correspond to land and ocean respectively.**

| Region | $DJF_L$ | $MAM_L$ | $ON_L$ | $DJF_O$ | $MAM_O$ | $ON_O$ |
|--------|---------|---------|--------|---------|---------|--------|
| West | 0.944 ±0.009 | 0.940 ±0.010 | 0.956 ±0.009 | 0.941 ±0.014 | 0.939 ±0.011 | 0.967 ±0.011 |
| East | 0.948 ±0.007 | 0.944 ±0.006 | 0.959 ±0.005 | 0.952 ±0.012 | 0.948 ±0.013 | 0.967 ±0.012 |

**Table 2: Difference between OSSA and cSSA. Suffixes L and O denote land and ocean respectively.**

| Region | $DJF_L$ | $MAM_L$ | $ON_L$ | $DJF_O$ | $MAM_O$ | $ON_O$ |
|--------|---------|---------|--------|---------|---------|--------|
| West | 0.031 | 0.026 | 0.051 | 0.009 | 0.005 | 0.015 |
| East | 0.038 | 0.043 | 0.053 | 0.005 | 0.005 | 0.006 |

**Table 3: Comparison of eSSA over the ocean with AERONET SSA from MCOH.**

| | DJF | MAM | ON |
|--------|-----|-----|-----|
| AERONET SSA | 0.958 ± 0.022 | 0.953 ± 0.013 | 0.951 ± 0.139 |
| eSSA | 0.956 ± 0.005 | 0.956 ± 0.159 | 0.971 ± 0.111 |



**Table 4: Mean slope along with its standard error of the linear regression fits created to identify zonal variation of columnar atmospheric heating rates imparted by aerosols for all seasons and sub-regions.**

| Season | Longitude (°E) | Slope (K d$^{-1}$ °longitude$^{-1}$) | | |
|---|---|---|---|---|
| | | SR1 | SR2 | SR3 |
| DJF | 65.5 - 77.5 | $6.3 \times 10^{-3} \pm 2.0 \times 10^{-3}$ | $3.4 \times 10^{-3} \pm 1.3 \times 10^{-3}$ | $1.4 \times 10^{-3} \pm 2.0 \times 10^{-4}$ |
| | 77.5 - 90.5 | $-4.2 \times 10^{-4} \pm 2.0 \times 10^{-3}$ | $-2.6 \times 10^{-3} \pm 4.7 \times 10^{-4}$ | $-1.8 \times 10^{-3} \pm 3.3 \times 10^{-4}$ |
| MAM | 65.5 - 77.5 | $3.3 \times 10^{-3} \pm 2.4 \times 10^{-3}$ | $8.0 \times 10^{-3} \pm 9.0 \times 10^{-4}$ | $1.9 \times 10^{-3} \pm 3.4 \times 10^{-4}$ |
| | 77.5 - 90.5 | $-6.0 \times 10^{-3} \pm 1 \times 10^{-3}$ | $-3.9 \times 10^{-3} \pm 8.3 \times 10^{-4}$ | $-2.6 \times 10^{-3} \pm 2.7 \times 10^{-4}$ |
| ON | 65.5 - 77.5 | $8.6 \times 10^{-3} \pm 2.0 \times 10^{-3}$ | $1.1 \times 10^{-2} \pm 9.8 \times 10^{-4}$ | $1.8 \times 10^{-3} \pm 2.7 \times 10^{-4}$ |
| | 77.5 - 90.5 | $-6.0 \times 10^{-3} \pm 1.7 \times 10^{-3}$ | $-1.1 \times 10^{-2} \pm 1.0 \times 10^{-3}$ | $-2.0 \times 10^{-3} \pm 3.2 \times 10^{-4}$ |