# Peer review of "Zonal variations of the vertical distribution of atmospheric aerosols over the Indian region and the consequent radiative effects"

_Atmospheric Chemistry and Physics, 2021_

## Author Comment (AC1)

**Author response to Reviewer #1 comments**

We sincerely thank the reviewer for the valuable comments. Based on the comments we received, careful modifications have been made to the manuscript. Our point-by-point response to the review comments are given below. The comments are marked in bold blue text and our responses are marked in normal black text below each comment.

**Reviewer #1**

**The paper is well written and appropriate for ACP. It provides a good overview of the aerosol conditions over the Indian Subcontinent. The CALIPSO data set is the basis for this and was carefully analysed.**

**Minor revisions are requested.**

We appreciate the summary evaluation and the positive comments.

**Abstract: The text is quite lengthy. I would appreciate a compact abstract, just 10-15 lines on methods and data, and a few key results. Many details are given in unnecessarily large detail. Such a detail description can be given in the summary section.**

Complied with. The shortened abstract is given below and is also shown enabling Track Changes in the revised version line 9, from P1.

The vertical structure of atmospheric aerosols over the Indian mainland and the surrounding oceans and its spatial distinctiveness and resultant atmospheric heating are characterized using long-term (2007 – 2020) satellite observations, assimilated aerosol single scattering albedo, and radiative transfer calculations. The results show strong, seasonally varying zonal gradients in the concentration and vertical extent of aerosols over the study region. Compared to the surrounding oceans, where the vertical extent of aerosols is confined within 3 km, the aerosol extinction coefficients extend to considerably higher altitudes over the mainland, reaching as high as 6 km during pre-monsoon and monsoon seasons. Longitudinally, the vertical extent is highest around 75°E and decreasing gradually towards either side of the study region, particularly over peninsular India. Particulate depolarization ratio profiles affirm the ubiquity of dust aerosols in western India from the surface to nearly 6 km. While the presence of low-altitude dust aerosols decreases further east, the high-altitude (above 4 km) dust layers remain aloft throughout the year with seasonal variations in its zonal distribution over north-western India. High-altitude (around 4 km) dust aerosols are observed over southern peninsular India and the surrounding oceans during the monsoon season. Radiative transfer calculations show that these changes in the vertical distribution of aerosols result in enhanced atmospheric heating at the lower altitudes during pre-monsoon, especially in the 2 – 3

km altitude range throughout the Indian region. These results have large implications for aerosol-radiation interactions in regional climate simulations.

**P2, line 54: Without these field campaigns, one would not know much about, e.g., lidar ratios, used in the CALIPSO data analysis. Such snapshot-like field campaigns are required to get a deep insight in aerosol properties. Without them, nobody would trust the 'larger picture' provided by the CALIPSO data sets.**

Thank you for the clarification. The following modification has been made in the revised manuscript in P2, line 47.

Such snapshot-like field campaigns provide deeper insight into the location-specific aerosol properties.

**P4, line 103-104: Are these lidar ratios from 23 to 70 sr in agreement with the lidar ratio observations realized during field campaigns such as INDOEX?**

Yes, they agree with the lidar ratio observed during INDOEX. CALIOP makes use of different lidar ratios to capture different aerosol species, ranging from 23 sr for clean marine aerosols to 70 sr for polluted continental aerosols at 532 nm wavelength (Kim et al., 2018). The measurements over Maldives showed lidar ratios between 60 to 90 sr (Ansmann et al., 2000). Franke et al., (2001) reported that high lidar ratios, reaching up to 110 sr, are associated to lofted pollution plumes, >70 sr is indicative of small absorbing aerosols, 30 to 36 sr for a mixture of marine and anthropogenic aerosols, and <30 sr for clean marine conditions. Müller et al., (2003) have reported lidar ratios over Maldives to be within 30 to 90 sr for a mixture of clean marine and clean continental aerosols. Franke et al., (2003) obtained lidar ratios between 30 to 100 sr at 532 nm for composite aerosols and between 50 to 80 sr for the absorbing aerosols advected from northern India.

**P4, l105-110: When using CALIPSO backscatter observations and MODIS AOT values one has the chance to get the column lidar ratios (AOD divided by the column backscatter). These values should be in harmony with the CALIPSO lidar ratios used in your data analysis.**

**Such studies are just indications that you did a lot in terms of quality assurance.**

Thank you for pointing this out. The following sentences have been added in the revised manuscript in P4, line 100.

The CALIOP aerosol extinction profiles are retrieved from the backscatter signals using lidar ratios as reported in Kim et al., (2018). As an additional quality check, we calculated the lidar ratios separately for a representative month (January 2010) by dividing MODIS AOD with the Level-2 CALIOP column-integrated aerosol backscattering coefficient. Mean lidar ratios of 34.1 sr, 42.6 sr, and 23.3 sr were observed respectively over SR1,

SR2 and SR3, which are in good agreement with the past lidar ratio observations surrounding the Indian region (Ansmann et al., 2000; Franke et al., 2001; 2003) and the lidar ratios employed in CALIOP. Furthermore, to overcome this limitation due to uncertainties in lidar ratios, the aerosol extinction coefficient profiles have been normalized using the combined Level-3 Version 6.1 average aerosol optical depth (AOD) from Moderate Resolution Imaging Spectroradiometer (MODIS) on board Aqua and Terra satellites (Wei et al., 2019).

**Page 6 is terrible with all the ASSA, OSSA; deltaSSA, eSSA. I had to write down all the abbreviations to get not lost.**

We are sorry for the inconvenience. We have replaced the abbreviations OSSA and eSSA with OMI SSA and extended SSA respectively in the entire manuscript and hope this would improve the readability.

**P10, l297: Do you know the papers of Hofer et al. (ACP 2017, 2020) on central Asian dust observations (Dushanbe, Tajikistan), and Hu et al. (ACP, 2021) on western China dust observation (Taklamakan area). These papers could be cited.**

**Hofer, J., Althausen, D., Abdullaev, S. F., Makhmudov, A. N., Nazarov, B. I., Schettler, G., Engelmann, R., Baars, H., Fomba, K. W., Müller, K., Heinold, B., Kandler, K., and Ansmann, A.: Long-term profiling of mineral dust and pollution aerosol with multiwavelength polarization Raman lidar at the Central Asian site of Dushanbe, Tajikistan: case studies, Atmos. Chem. Phys., 17, 14559–14577, https://doi.org/10.5194/acp-17-14559-2017, 2017.**

**Hofer, J., Ansmann, A., Althausen, D., Engelmann, R., Baars, H., Fomba, K. W., Wandinger, U., Abdullaev, S. F., and Mahmudur, A. N.: Optical properties of Central Asian aerosol relevant for spaceborne lidar applications and aerosol typing at 355 and 532 nm, Atmos. Chem. Phys., 20, 9265–9280, https://doi.org/10.5194/acp-20-9265-2020, 2020.**

**Hu, Q., Wang, H., Goloub, P., Li, Z., Veselovskii, I., Podvin, T., Li, K., and Korenskiy, M.: The characterization of Taklamakan dust properties using a multiwavelength Raman polarization lidar in Kashi, China, Atmos. Chem. Phys., 20, 13817–13834, https://doi.org/10.5194/acp-20-13817-2020, 2020.**

Thank you. Complied with in P11, line 322 in the revised manuscript.

**Fig. 2: I would appreciate if OSSA and cSSA would be explained in the figure caption….**

Complied with. Modified figure caption in P27, line 791 is as shown below:

Figure 2: Spatial map of OMI SSA (panels a, b, and c) and cSSA (combined SSA; panels d, e, and f). The left, middle, and right panels represent the seasons DJF, MAM, and ON respectively.

**Fig.3: Please state in the caption that these observations are taken by CALIOP. What shows the dashed line in Fig.3c?**

Complied with. Dotted yellow line in Fig. 3c illustrates the eastward gradient in the extent of vertical intrusion of aerosols. The modified figure caption is shown in P28, line 797 as shown below:

Figure 3: Zonal variation of the CALIOP aerosol extinction coefficient ($k_{ext}$) profiles for SR1 (top panel), SR2 (middle panel), and SR3 (bottom panel) sub-regions. Each column corresponds to a particular season, as marked above them. Dotted yellow line in Fig. 3c illustrates the eastward gradient in the extent of vertical intrusion of aerosols.

**Fig.4, the PDR values are derived from the respective VDR values by using the particle backscatter coefficient profiles. So the PDR profiles are uncertain. How trustworthy they are… should be discussed! For example, the red PDR fields (4b, 4c) show values of 30-40% depolarization ratio as THREE MONTH MEAN VALUES. Such high values cannot be explained by pure mineral dust. Desert dust produces 30% (so yellow colors), and at extreme conditions, close to dust sources, 35% may happen, but only in some cases.**

The colour scheme used had some problems. The figure (Fig. 4) has now been remade in the revised manuscript and shows that PDR values above 0.35 to be seldom observed over the study region, and generally staying within 0.35. Prijith et al., (2016) have reported similar values of seasonal mean PDR over the Indian region, staying within 0.35, and occasionally going up to even 0.5.

**Fig.5: cSSA, OSSA should be explained in the caption.**

Complied with. The modified figure caption in P30, line 807 is shown below:

Figure 5: Zonal variation of cSSA (combined SSA) and OMI SSA in SR1, SR2, and SR3 during DJF, MAM, and ON seasons. The rows from top to bottom correspond to sub-regions SR1, SR2, and SR3 respectively.

**References**

- Ansmann, A., Althausen, D., Wandinger, U., Franke, K., Müller, D., Wagner, F., & Heintzenberg, J.: Vertical profiling of the Indian aerosol plume with six-wavelength lidar during INDOEX: A first case study, Geophysical Research Letters, 27(7), 963-966, 2000.

- Franke, K., Ansmann, A., Müller, D., Althausen, D., Wagner, F., and Scheele, R.: One-year observations of particle lidar ratio over the tropical Indian Ocean with Raman lidar. Geophysical Research Letters, 28(24), 4559-4562, 2001.

- Franke, K., Ansmann, A., Müller, D., Althausen, D., Venkataraman, C., Reddy, M.S., Wagner, F. and Scheele, R..: Optical properties of the Indo-Asian haze layer over the tropical Indian Ocean. Journal of Geophysical Research: Atmospheres, 108(D2), 2003.

- Hofer, J., Althausen, D., Abdullaev, S. F., Makhmudov, A. N., Nazarov, B. I., Schettler, G., Engelmann, R., Baars, H., Fomba, K. W., Müller, K., Heinold, B., Kandler, K., and Ansmann, A.: Long-term profiling of mineral dust and pollution aerosol with multiwavelength polarization Raman lidar at the Central Asian site of Dushanbe, Tajikistan: case studies, Atmospheric Chemistry and Physics., 17, 14559–14577, https://doi.org/10.5194/acp-17-14559-2017, 2017.

- Hofer, J., Ansmann, A., Althausen, D., Engelmann, R., Baars, H., Fomba, K. W., Wandinger, U., Abdullaev, S. F., and Mahmudur, A. N.: Optical properties of Central Asian aerosol relevant for spaceborne lidar applications and aerosol typing at 355 and 532 nm, Atmospheric Chemistry and Physics., 20, 9265–9280, https://doi.org/10.5194/acp-20-9265-2020, 2020.

- Hu, Q., Wang, H., Goloub, P., Li, Z., Veselovskii, I., Podvin, T., Li, K., and Korenskiy, M.: The characterization of Taklamakan dust properties using a multiwavelength Raman polarization lidar in Kashi, China, Atmospheric Chemistry and Physics., 20, 13817–13834, https://doi.org/10.5194/acp-20-13817-2020, 2020.

- Kim, M.H., Omar, A.H., Tackett, J.L., Vaughan, M.A., Winker, D.M., Trepte, C.R., Hu, Y., Liu, Z., Poole, L.R., Pitts, M.C. and Kar, J.: The CALIPSO version 4 automated aerosol classification and lidar ratio selection algorithm. Atmospheric Measurement Techniques, 11(11), 6107-6135, 2018.

- Müller, D., K. Franke, A. Ansmann, and D. Althausen (2003), Indo-Asian pollution during INDOEX: Microphysical particle properties and single-scattering albedo inferred from multiwavelength lidar observations, J. Geophys. Res., 108(D19), 4600, doi:10.1029/2003JD003538.

- Prijith, S. S., Babu, S. S., Lakshmi, N. B., Satheesh, S. K., and Moorthy, K. K.: Meridional gradients in aerosol vertical distribution over Indian Mainland: Observations and model simulations, Atmospheric environment, 125, 337-345, 2016.

- Wei, J., Peng, Y., Guo, J., and Sun, L.: Performance of MODIS Collection 6.1 Level 3 aerosol products in spatial-temporal variations over land, Atmospheric Environment, 206, 30-44, 2019.

---

## Author Comment (AC2)

**Author response to Reviewer #2 comments**

We sincerely thank the reviewer for the valuable comments. Based on the comments we received, careful modifications have been made to the manuscript. Our point-by-point response to the review comments are given below. The comments are marked in bold blue text and our responses are marked in normal black text below each comment.

**Reviewer #2**

**Zonal variations of the vertical distribution of atmospheric aerosols over the Indian region and the consequent radiative effects Kala et al.,**

**A compilation of vertical profile and horizontal data for atmospheric aerosol properties (SSA, extinction coefficients and depolarization ratios). This group has produced several papers that have similar flavour to this one, primarily taking data and doing radiative forcing calculations using SBDART. The 'novelty' here may be the use of CALIPSO data to build vertical profiles that seem to have been rescaled using surface observations from the various field studies conducted around India, including the ICARB. I don't have any big issues with the paper, except that it doesn't offer anything new in terms of analysis/modelling. The two major concerns would be.**

**1.        There is not much ground validation for the data in terms of comparisons with some ground based or aircraft data collection performed by the team.**

A detailed comparison of our results with previous ground-based and aircraft-based lidar observations has been added to the revised manuscript in P10, line 291 as shown below.

Our $k_{ext}$ values are in good agreement with the results of Satheesh et al., (2006) using micro pulse lidar, where the presence of aerosol layers aloft (within $1 - 2$ km) during DJF over the urban region of Bengaluru (13.01°N, 77.34°E) was reported. Satheesh et al., (2008) reported the large concentration of aerosols within $2 - 5$ km over the inland regions of peninsular India during the Integrated Campaign for Aerosols, gases and Radiation Budget (ICARB). They highlighted the decrease in the vertical extent (from 5 km over Central India to 1 km in the Northern Indian Ocean) and concentration of aerosols (a reduction in the maximum value of $k_{ext}$ in the atmospheric column from 0.4 $km^{-1}$ in Central India to 0.2 $km^{-1}$ in Northern Indian Ocean) as we move meridionally away from the land into the surrounding oceans. Airborne lidar measurements during ICARB reported two Eastern coastal regions in peninsular India, namely Bhubaneshwar and Chennai, to have the vertical extent of $k_{ext}$ as 3 km and 5 km respectively during MAM (Satheesh et al., 2009a). The maximum values of $k_{ext}$ within the atmospheric column were observed be 0.3 $km^{-1}$ and 0.35 $km^{-1}$ respectively. Meanwhile, the southern coastal region Trivandrum had the vertical extent of $k_{ext}$ to be 1.5 km and having a maximum value of 0.15 $km^{-1}$. It should be noted that all these stations had the maximum value of $k_{ext}$ to be at altitudes above 2 km and not close to the surface. Satheesh et al.,

(2009a; 2009b) showed how the vertical extent and magnitude of $k_{ext}$ decreases away from the coasts of three regions in the Indian peninsula. The contribution of aerosols above 3 km to columnar optical depth (AOD) was observed to decrease with increasing distance off the shorelines for Chennai whereas it remained more or less independent of distance for Bhubaneshwar. Lidar measurements by Moorthy et al. (2008) reported the presence of high-altitude aerosols even 400 km away from the coasts of Bay of Bengal, with a decreasing vertical extent and concentration (with a maximum value of 0.15 $km^{-1}$ in the $k_{ext}$ profile) away from the Chennai coast. Vaishya et al., (2018) made use of aircraft observations to study the vertical profiles of $k_{ext}$ over Western India (Jodhpur), Central India (Varanasi), and Eastern India (Bhubaneshwar) during the South West Asian Aerosol Monsoon Interactions (SWAAMI) campaign conducted along with the Regional Aerosol Warming Experiment (RAWEX) campaign in June 2016. Peak value in the $k_{ext}$ profile was reported by them to be maximum (0.2 $km^{-1}$) over Central India and reducing to either side to attain peak $k_{ext}$ values of ~0.1 $km^{-1}$ over Western and Eastern India. Manoj et al., (2020) reported the presence of aerosols above 2 km during the onset of monsoon over Northern India (Lucknow), Central India (Nagpur), North-Western India (Jaipur), Arabian Sea and Bay of Bengal during the SWAAMI aircraft campaign. These observations are in excellent agreement with our observations on the seasonality and zonal gradients in $k_{ext}$ shown in Fig. 3.

**2.      The 'correction' of the ASSA over the ocean using for profiles uses the OSSA extended over the ocean and obtain a regression factor that was applied to ASSA. This seems arbitrary in some sense. Why not use a physics informed method that uses the differences in temperature profiles, water vapor profiles or PBL heights between the coastal and overland regions to inform the corrections?**

The assimilated SSA is constructed from the gridded columnar Aerosol Optical Depth (AOD) and Aerosol Absorption Optical Depth (AAOD); both constructed by assimilating ground-based aerosol measurements with gridded satellite-retrieved products (Pathak et al., 2019) and employing data assimilation techniques that take into account the meteorological (Planetary Boundary Layer height) and topographical factors (elevation) with their inherent spatio-temporal variation. The assimilated Single Scattering Albedo (SSA) demonstrates more accurate columnar SSA values with substantially smaller uncertainties, vis-a-vis satellite-retrieved SSA. However, the assimilated SSA are available only over land regions due to the lack of long-term aerosol measurements over the oceanic regions and the fewer number of observatories. Therefore, we have extended the assimilated SSA over the oceanic regions by using the spatial variation demonstrated by satellite retrieved SSA, which exhibits close agreement with large-scale spatial patterns in assimilated SSA over the inland regions. This method is implicitly considering the local meteorological factors (through the assimilated aerosol products) as well as the realistic spatial distribution of SSA (through the spatial variation by satellite SSA) over the oceanic region obtained from Ozone Monitoring Instrument (OMI). This is based on the rationale that the SSA has higher variability over the land than over the ocean. Satheesh et al., (2010) have shown a higher gradient in SSA over the Arabian Sea and Bay of Bengal is in the north-south direction. Hence, taking this into account, this method employs the extension with the variation of SSA with respect to longitude for every latitude. The extended SSA is compared with previous reports of SSA over the oceanic regions in Table RC1. It can be observed that our extended SSA data values are in better agreement with the previous observations.

**Table RC1: Comparison of SSA between previous in-situ observations with the OMI SSA and extended assimilated SSA reported in the present study.**

| Sl. No. | Region | SSA observations | | | Period | Reference |
|---|---|---|---|---|---|---|
| | | **Past** | **Present** | **OMI** | | |
| 1 | Bay of Bengal (BoB) | 0.93±0.03 | 0.94±0.02 | 0.95±0.01 | April 1999 | Nair et al., (2008) |
| 2 | Kaashidhoo | 0.9 | 0.95 | 0.96 | Feb-Mar 1998 | Satheesh et al., (1999) |
| 3 | Hanimaadhoo | 0.93±0.02 | 0.95 | 0.96 | Nov-Dec 2009 | Corrigan et al., (2006) |
| 4 | BoB | 0.93±0.01 | 0.94±0.02 | 0.95±0.01 | Mar-Apr 2006 | Kedia et al., (2010) |
| 5 | AS | 0.96±0.01 | 0.94±0.01 | 0.94±0.01 | Apr-May 2006 | Kedia et al., (2010) |
| 6 | Kavaratti | 0.91±0.01 | 0.94 | 0.96 | Mar-May 2012 | Patel et al., (2015) |
| 7 | AS | 0.92±0.01 | 0.93+0.01 | 0.95±0.01 | Mar-Apr 2003 | Moorthy et al., (2005) |
| 8 | BoB | 0.93 | 0.93 | 0.94 | Dec-Apr 2001 | Ramachandran et al., (2005) |
| 9 | BoB | 0.9 | 0.94±0.01 | 0.95±0.01 | Feb 2003 | Ganguly et al., (2005) |

**Beyond these two, the manuscript badly needs a comparative evaluation with some model simulations. It is hard to get a sense to understand how this will feedback into improving models (regional and global). There are several runs performed as part of the CMIP6 with GCMs of various resolution and model output from the (AerChemMIP) for example. These should be accessible; how does this dataset compare with these simulations. There is a lot of qualitative description of mixing and gradients that are driven by dynamics. Using a model result to put these in context would be essential and making all the discussion more concrete. Without an accompanying model evaluation, the added value of this product to literature is questionable.**

A detailed comparison between our results and AerChemMIP model simulations has been carried out as discussed below and has been added as a supplementary material (P1, line 13) for the revised manuscript.

A detailed comparison of our results with Aerosols and Chemistry Model Intercomparison Project (AerChemMIP) under the Coupled Model Inter Comparison Project (CMIP6) model from Meteorological Research Institute – Earth System Model (MRI-ESM) 2.0 (Yukimoto et al., 2019) is carried out for aerosol extinction coefficient ($k_{ext}$) and dust AOD. The present study uses data during the time period 2006–2020, and for a comparison, the scenarios

considered in AerChemMIP6 are Historical Sea Surface Temperature (HistSST; 2006-2014) and Shared Socioeconomic Pathway (SSP3-7.0; 2015-2020). These two scenarios were chosen to match the time-period between AerChemMIP6 and the present study. The rationale for using these two scenarios for comparison is that they are baseline simulations consistent with observations (Collins et al., 2017; Lund et al., 2019). A short description of the two scenarios is given below:

1) HistSST scenario (Meinshausen et al., 2017): These simulations impose changes that are consistent with observations. The model performances are evaluated against the present climate and observed climate change.

2) SSP3-7.0 scenario (O'Neill et al., 2014): These are gap-filling simulations in the CMIP5 forcing pathways and forms baseline forcing levels for several (unmitigated) scenarios.

Fig. S1 shows the AerChemMIP6 model simulation of $k_{ext}$ for the same location and time period as in the present study (Fig. 3). The results are consistent with our results and exhibit a zonal gradient from the west to the east, even the small hump in the middle being reproduced. The increase in vertical extent and magnitude of $k_{ext}$ over the west during JJAS is also comparable. Even though the model simulations are in good agreement with the zonal gradients and the magnitudes of $k_{ext}$ in Fig. 3 on a larger scale, our results reveal that the AerChemMIP6 simulations are underestimates over finer spatial scales. The high $k_{ext}$ values and its vertical extent in the west (see Fig. 3) around the monsoon season is attributed to the long-range transport of dust aerosols (Banerjee et al., 2019; 2021). The dust AOD values from AerChemMIP6, shown in Fig. S2, also show high values over the west during MAM and JJAS, particularly over SR1. This agrees with our attribution of the dust influence to high $k_{ext}$ over the Indian region (especially over the west) during JJAS and MAM seasons, as shown in Fig. 3 – 4. Our results will therefore prove useful in improving the regional climate model simulations.

[Figure]

**Figure S1: Zonal variation of the aerosol extinction coefficient ($k_{ext}$) (MRI-ESM2 model simulations) profiles for SR1 (top panel), SR2 (middle panel), and SR3 (bottom panel) sub-regions. Each column corresponds to a particular season, as marked above them.**

[Figure]

**Figure S2: Zonal variation of the dust AOD (MRI-ESM2 model simulations) for SR1 (top panel), SR2 (middle panel), and SR3 (bottom panel) sub-regions. Each column corresponds to a particular season, as marked above them.**

**Some Specific Comments:**

**Line 45: Feng et al., 2016 did a detailed evaluation of the radiative forcing due to differences in land and ocean vertical profiles using MPLNet, CALIPSO and WRF-CHEM (doi:10.5194/acp-16-247-2016) and seems highly relevant to work discussed here. How do the calculations on radiative forcing performed here differ or similar to that discussed in that publication?**

A comparison with the radiative forcing results of Feng et al. (2016) was carried out and the following section has been added as a supplementary material (P3, line 44) for the revised manuscript.

The present work utilizes observational datasets like CALIOP aerosol extinction coefficient, Moderate Resolution Imaging Spectroradiometer (MODIS) AOD and assimilated SSA to evaluate atmospheric radiative forcing using Santa Barbara DISORT Atmospheric Radiative Transfer (SBDART) model. Feng et al., (2016) used the Rapid Radiative Transfer Model (RRTM) for radiative transfer calculations in the Weather Research Forecasting with Chemistry (WRF-Chem) model. For comparison, the shortwave aerosol-induced atmospheric heating rate (dT/dt) have been estimated using our data sets and SBADART for the same region and time period (55–95˚E and 0–36˚N, March 2012) as in Feng et al., (2016). These results are compared with the control runs for shortwave dT/dt simulations shown in Fig. 4a (land) and Fig. 4d (ocean) in Feng et al., (2016), and are shown in Fig. S3 below. The magnitudes of dT/dt are higher in the present work as compared to Feng et al., (2016), but the vertical variations are more or less similar. The mismatch in the magnitudes of dT/dt is understandable because of two reasons: (1) Our dT/dt calculations make use of a gamut of realistic observations as inputs while the model makes use of simulated parameters as inputs, (2) There is a large underestimation of $k_{ext}$ in the model simulations as

compared to the observations (as high as a factor of four), as can be seen in Fig. 2 of Feng et al., (2016). This comparison further elucidates the importance of our results for improving regional climate simulations.

[Figure]

**Figure S3: Vertical variation of shortwave aerosol-induced atmospheric heating rate (dT/dt) profiles over (a) Indian mainland and (b) oceanic regions.**

**Line 204:  What dynamics are of importance here? Synoptic, mesoscale or boundary layer?**

The dynamics of the atmospheric boundary layer governs the concentration and vertical distribution of aerosols in the lower altitudes whereas mesoscale and synoptic scales have a role in deciding the high-altitude aerosol concentration, through the long-range transported and elevated aerosol layers. Hence, three processes, mainly the long-range transport, accumulation, and dispersion of the aerosols regulate the zonal gradients discussed in the present study (Prijith et al., 2013; Ratnam et al., 2018; 2021).

**Table 4: How do these heating rates compare to those being calculated by GCMs and models from AEROChemMIP?**

[Figure]

**Figure RC4: Zonal variation of GFDL-ESM4 model simulated aerosol-induced atmospheric heating rate profiles (dT/dt) for SR1 (top panel), SR2 (middle panel), and SR3 (bottom panel) sub-regions. Each column corresponds to a particular season, as marked above them.**

The dT/dt values from Geophysical Fluid Dynamics Laboratory – Earth System Model (GFDL-ESM4) model for HistSST experiment is shown in Figure RC4. While in general, the zonal variation and the vertical extent of dT/dt agree with our findings (see Fig. 6), the absolute magnitudes do not match, which is quite understandable, as the model simulations do not incorporate the realistic vertical distribution or SSA of the aerosols. dT/dt is higher in GFDL-ESM4 compared to our work (especially in the lower altitudes) possibly due to the underestimation of SSA in GFDL-ESM4, as shown in Mallet et al., (2021), who has categorized GFDL-ESM4 as 'C-' group [i.e., SSA has a negative bias compared to Aerosol Robotic Network (AERONET) measurements]. Hence, GFDL-ESM4 would have lesser SSA (compared to the observations) as input to the radiative transfer calculations, which may be the reason for their overestimation of dT/dt.

**References**

- Banerjee, P., Satheesh, S. K., Moorthy, K. K., Nanjundiah, R. S., and Nair, V. S.: Long-range transport of mineral dust to the northeast Indian Ocean: Regional versus remote sources and the implications. Journal of Climate, 32(5), 1525-1549, 2019.
- Banerjee, P., Satheesh, S. K., and Moorthy, K. K.: Is the Atlantic Ocean driving the recent variability in South Asian dust? Atmospheric Chemistry and Physics, 21(23), 17665-17685, 2021.

- Collins, W.J., Lamarque, J.F., Schulz, M., Boucher, O., Eyring, V., Hegglin, M.I., Maycock, A., Myhre, G., Prather, M., Shindell, D. and Smith, S.J.: AerChemMIP: quantifying the effects of chemistry and aerosols in CMIP6. Geoscientific Model Development, 10(2), 585-607, 2017.

- Corrigan, C. E., Ramanathan, V., and Schauer, J. J.: Impact of monsoon transitions on the physical and optical properties of aerosols. Journal of Geophysical Research: Atmospheres, 111(D18), 2006.

- Feng, Y., Kotamarthi, V. R., Coulter, R., Zhao, C., and Cadeddu, M.: Radiative and thermodynamic responses to aerosol extinction profiles during the pre-monsoon month over South Asia. Atmospheric Chemistry and Physics, 16(1), 247-264, 2016.

- Ganguly, D., Jayaraman, A., and Gadhavi, H.: In situ ship cruise measurements of mass concentration and size distribution of aerosols over Bay of Bengal and their radiative impacts. Journal of Geophysical Research: Atmospheres, 110(D6), 2005.

- Kedia, S., Ramachandran, S., Kumar, A., and Sarin, M. M.: Spatiotemporal gradients in aerosol radiative forcing and heating rate over Bay of Bengal and Arabian Sea derived on the basis of optical, physical, and chemical properties. Journal of Geophysical Research: Atmospheres, 115(D7), 2010.

- Lund, M. T., Myhre, G., and Samset, B. H.: Anthropogenic aerosol forcing under the Shared Socioeconomic Pathways. Atmospheric Chemistry and Physics, 19(22), 13827-13839, 2019.

- Manoj, M. R., Satheesh, S. K., Moorthy, K. K., and Coe, H.: Vertical profiles of submicron aerosol single scattering albedo over the Indian region immediately before monsoon onset and during its development: research from the SWAAMI field campaign. Atmospheric Chemistry and Physics, 20(6), 4031-4046, 2020.

- Mallet, M., Nabat, P., Johnson, B., Michou, M., Haywood, J. M., Chen, C., and Dubovik, O.: Climate models generally underrepresent the warming by Central Africa biomass-burning aerosols over the Southeast Atlantic. Science advances, 7(41), 2021.

- Moorthy, K. K., Babu, S. S., and Satheesh, S. K.: Aerosol characteristics and radiative impacts over the Arabian Sea during the intermonsoon season: Results from ARMEX field campaign. Journal of the Atmospheric Sciences, 62(1), 192-206, 2005.

- Moorthy, K. K., Satheesh, S. K., Babu, S. S., and Dutt, C. B. S.: Integrated campaign for aerosols, gases and radiation budget (ICARB): an overview. Journal of Earth System Science, 117(1), 243-262, 2008.

- Meinshausen, M., Vogel, E., Nauels, A., Lorbacher, K., Meinshausen, N., Etheridge, D.M., Fraser, P.J., Montzka, S.A., Rayner, P.J., Trudinger, C.M. and Krummel, P.B.: Historical greenhouse gas concentrations for climate modelling (CMIP6). Geoscientific Model Development, 10(5), 2057-2116, 2017.

- Nair, V. S., Babu, S. S., and Moorthy, K. K.: Spatial distribution and spectral characteristics of aerosol single scattering albedo over the Bay of Bengal inferred from shipborne measurements. Geophysical Research Letters, 35(10), 2008.

- O'Neill, B.C., Kriegler, E., Riahi, K., Ebi, K.L., Hallegatte, S., Carter, T.R., Mathur, R. and van Vuuren, D.P.: A new scenario framework for climate change research: the concept of shared socioeconomic pathways. Climatic change, 122(3), 387-400, 2014.

- Patel, P., and Shukla, A. K.: Aerosol optical properties over marine and continental sites of India during pre-monsoon season. Current Science, 666-676, 2015.

- Pathak, H. S., Satheesh, S. K., Nanjundiah, R. S., Moorthy, K. K., Lakshmivarahan, S., and Babu, S. N. S.: Assessment of regional aerosol radiative effects under the SWAAMI campaign–Part 1: Quality-enhanced

estimation of columnar aerosol extinction and absorption over the Indian subcontinent, Atmospheric Chemistry and Physics, 19, 11865-11886, 2019.

- Prijith, S.S., Aloysius, M. and Mohan, M.: Global aerosol source/sink map. Atmospheric Environment, 80, pp.533-539, 2013.

- Ramachandran, S.: Aerosol radiative forcing over Bay of Bengal and Chennai: Comparison with maritime, continental, and urban aerosol models. Journal of Geophysical Research: Atmospheres, 110(D21), 2005.

- Ratnam, M. V., Prasad, P., Roja Raman, M., Ravikiran, V., Bhaskara Rao, S. V., Krishna Murthy, B. V., and Jayaraman, A.: Role of dynamics on the formation and maintenance of the elevated aerosol layer during monsoon season over south-east peninsular India, Atmospheric Environment, 188, 43-49, https://doi.org/10.1016/j.atmosenv.2018.06.023, 2018.

- Ratnam, M. V., Prasad, P., Raj, S. T. A., Raman, M. R., and Basha, G.: Changing patterns in aerosol vertical distribution over South and East Asia, Scientific Reports, 11, 1-11, 2021.

- Satheesh, S. K., Ramanathan, V., Xu Li-Jones, Lobert, J. M., Podgorny, I. A., Prospero, J. M., Holben, B. N., and Loeb. N. G.: A model for the natural and anthropogenic aerosols over the tropical Indian Ocean derived from Indian Ocean Experiment data. Journal of Geophysical Research: Atmospheres, 104(D22), 27421-27440, 1999.

- Satheesh, S. K., Vinoj, V., and Moorthy, K. K.: Vertical distribution of aerosols over an urban continental site in India inferred using a micro pulse lidar. Geophysical Research Letters, 33(20), 2006.

- Satheesh, S. K., Moorthy, K. K., Babu, S. S., Vinoj, V., and Dutt, C. B. S.: Climate implications of large warming by elevated aerosol over India. Geophysical Research Letters, 35(19), 2008.

- Satheesh, S. K., Moorthy, K. K., Suresh Babu, S., Vinoj, V., Nair, V. S., Naseema Beegum, S., Dutt, C. B. S., Alappattu, D. P., and Kunhikrishnan, P. K.: Vertical structure and horizontal gradients of aerosol extinction coefficients over coastal India inferred from airborne lidar measurements during the Integrated Campaign for Aerosol, Gases and Radiation Budget (ICARB) field campaign. Journal of Geophysical Research: Atmospheres, 114(D5), 2009a.

- Satheesh, S. K., Vinoj, V., Suresh Babu, S., Moorthy, K. K., and Nair, V. S.: Vertical distribution of aerosols over the east coast of India inferred from airborne LIDAR measurements. Annales geophysicae (Vol. 27, No. 11, pp. 4157-4169). Copernicus GmbH, 2009b.

- Satheesh, S. K., Vinoj, V., and Moorthy, K. K.: Assessment of aerosol radiative impact over oceanic regions adjacent to Indian subcontinent using multi satellite analysis. Advances in Meteorology, 2010.

- Vaishya, A., Babu, S. N. S., Jayachandran, V., Gogoi, M. M., Lakshmi, N. B., Moorthy, K. K., and Satheesh, S. K.: Large contrast in the vertical distribution of aerosol optical properties and radiative effects across the Indo-Gangetic Plain during the SWAAMI–RAWEX campaign. Atmospheric Chemistry and Physics, 18(23), 17669-17685, 2018.

- Yukimoto, S., Kawai, H., Koshiro, T., Oshima, N., Yoshida, K., Urakawa, S., Tsujino, H., Deushi, M., Tanaka, T., Hosaka, M. and Yabu, S.: The Meteorological Research Institute Earth System Model version 2.0, MRI-ESM 2.0: Description and basic evaluation of the physical component. Journal of the Meteorological Society of Japan. Ser. II, 2019.

---

## Author Response (AR2)

**Author response to review comments**

We sincerely thank the Editor and the reviewers for their valuable comments. Based on the comments we received, careful modifications have been made to the manuscript. Our point-by-point response to the review comments are given below. The comments are marked in bold blue text and our responses are marked in normal black text below each comment. The changes made in the revised manuscript are also provided.

**Comments**

**Dear Nair Krishnan Kala,**

**please respond to the comments of one referee who is requesting the following minor changes:**

We express our sincere thanks to the Editor for the email notifying the review comments and decision for our manuscript.

**Please cite the material in the supporting document in the main manuscript at appropriate locations. As it reads this hasn't been done and you should link to supporting material some of the arguments you made in the manuscript in reference to zonal gradients and comparisons to previous work.**

Complied with. Thank you for the suggestion.

The following sentences have been added to page 13 at line number 389 (Sect 3.3) in the revised manuscript.

"Aerosols and Chemistry Model Intercomparison Project (AerChemMIP) simulations of $k_{ext}$ for the same location and time period, presented in detail in supplementary section (S1), revealed a similar zonal gradient with an increasing gradient from the west to the east, a maximum in the centre and a reduction thereafter towards the east. Even though the model simulations are in good agreement with the zonal gradients and the magnitudes of $k_{ext}$ in Fig. 3 on a larger scale, our results reveal that the AerChemMIP6 simulations are underestimated over finer spatial scales."

The following sentences have been added to page 14 at line number 434 (Sect 3.4) in the revised manuscript.

"A previous study by Feng et al. (2016) has shown the profiles of dT/dt estimates over Indian landmass and oceanic region separately using Rapid Radiative Transfer Model (RRTM) for radiative transfer calculations in the WRF-Chem model. dT/dt in their study showed higher values over land than over the ocean, with land exhibiting an exponential decrease vertically from high values at the surface. A detailed comparison of our results with Feng et al. (2016) can be found in the supplementary section (S2). These observations are consistent with the present work."